# Reconstructing the history of founder events using genome-wide patterns of allele sharing across individuals

Rémi Tournebize[1,2¤]*, Gillian Chu[3], Priya Moorjani[1,2]*

**1** Department of Molecular and Cell Biology, University of California, Berkeley, California, United States of America, **2** Center for Computational Biology, University of California, Berkeley, California, United States of America, **3** Department of Electrical Engineering and Computer Science, University of California, Berkeley, California, United States of America

¤ Current address: Instituto Gulbenkian de Ciência, Oeiras, Portugal
* remi.tournebize@gmail.com (RT); moorjani@berkeley.edu (PM)

**Data Availability Statement:** The data underlying the results presented in the study are available from the Reich lab (Allen DNA Resource): https://reich.hms.harvard.edu/allen-ancient-dna-resource-aadr-downloadable-genotypes-present-day-and-

## Abstract

Founder events play a critical role in shaping genetic diversity, fitness and disease risk in a population. Yet our understanding of the prevalence and distribution of founder events in humans and other species remains incomplete, as most existing methods require large sample sizes or phased genomes. Thus, we developed *ASCEND* that measures the correlation in allele sharing between pairs of individuals across the genome to infer the age and strength of founder events. We show that *ASCEND* can reliably estimate the parameters of founder events under a range of demographic scenarios. We then apply *ASCEND* to two species with contrasting evolutionary histories: ∼460 worldwide human populations and ∼40 modern dog breeds. In humans, we find that over half of the analyzed populations have evidence for recent founder events, associated with geographic isolation, modes of sustenance, or cultural practices such as endogamy. Notably, island populations have lower population sizes than continental groups and most hunter-gatherer, nomadic and indigenous groups have evidence of recent founder events. Many present-day groups—including Native Americans, Oceanians and South Asians—have experienced more extreme founder events than Ashkenazi Jews who have high rates of recessive diseases due their known history of founder events. Using ancient genomes, we show that the strength of founder events differs markedly across geographic regions and time—with three major founder events related to the peopling of Americas and a trend in decreasing strength of founder events in Europe following the Neolithic transition and steppe migrations. In dogs, we estimate extreme founder events in most breeds that occurred in the last 25 generations, concordant with the establishment of many dog breeds during the Victorian times. Our analysis highlights a widespread history of founder events in humans and dogs and elucidates some of the demographic and cultural practices related to these events.

ancient-dna-data The method is available on a public repository: https://github.com/sunyatin/ASCEND.

**Funding:** RT was supported by the UC Departmental Startup funds awarded to PM. PM has support from the Burroughs Wellcome Fund Careers at the Scientific Interface, Sloan Research Fellowship, and NIH R35GM142978. The funders had no role in study design, data collection and analysis, decision to publish, or preparation of the manuscript.

**Competing interests:** The authors have declared that no competing interests exist.

## Author summary

A founder event occurs when small numbers of ancestral individuals give rise to a large fraction of the population. Founder events reduce genetic variation and increase the risk of recessive diseases. Despite their importance in evolutionary and disease studies, we still only have a limited comprehension of their prevalence and properties in humans and other species, as most existing methods require large sample sizes or phased genomes. Here, we present a flexible method, *ASCEND*, to infer the timing and the strength of founder events that is suitable for sparse datasets with few samples or limited coverage. *ASCEND* provides reliable estimates across a wide range of demographic scenarios. By applying it to data from two species (humans and dogs), we document a widespread history of recent founder events in both species and provide insights about the demographic processes related to these events. Our analysis helps to identify groups with strong founder events that should be prioritized for future studies as they offer a unique opportunity for biological discovery and reducing disease burden through mapping of recessive disease-causing genes and pathways, as previously shown in studies of Ashkenazi Jews and Finns.

## Introduction

A founder event occurs when a new population is formed by a subset of individuals from a larger group or when the original population goes through a reduction in size due to a bottleneck [1]. Founder events have played a critical role in shaping genetic diversity in many species, including humans. For instance, anatomically modern humans spread worldwide in the past $\sim$ 50,000-100,000 years, following periods of successive bottlenecks and mixtures [2]. Many human populations have further undergone severe founder events in the recent past (past hundreds of generations) due to geographical isolation (e.g., Finns [3]) or historical migrations (e.g., Roma [4]) or cultural practices (e.g., Amish [5] and Ashkenazi Jews [6]).

Founder events reduce genetic variation in a population, decrease the efficacy of selection to remove deleterious variants, and increase the risk of recessive diseases [1]. Understanding the history of founder events can thus be helpful for learning about the cultural and demographic events leading to population bottlenecks, and importantly, for mapping functional and disease variants. Gene mapping efforts in founder populations—including Ashkenazi Jews, Finns, Amish, and French Canadians—have resulted in the discovery of numerous disease-causing mutations in each group [7] and refined our understanding of disease architectures [8].

Despite the importance of founder events in evolutionary and disease studies, we still only have a limited comprehension of their number, tempo, and strength in humans and other species. Characterizing the timing and strength of founder events is the first step towards improving our understanding of the impact of founder events on neutral and deleterious genetic variation. In particular, the estimated timing of the founder event (referred to as *founder age*, henceforth) can inform us about the cultural or environmental factors underlying the founder events. Further, it offers insights about the expected length of genomic segments that are inherited identical-by-descent (IBD) among individuals in the population. The strength of a founder event measured as the reduction in population size due to the bottleneck is informative about the probability of fixation of alleles, including deleterious and disease-associated variants [9]. Together, these parameters can reveal the evolutionary history and impact of founder events in shaping genetic diversity and disease risk.

There are two main classes of methods currently available for characterizing founder events: polymorphism-based and haplotype-based approaches. Polymorphism-based approaches leverage the observed patterns of genetic variation, either by studying the density of heterozygous sites in a region (e.g., PSMC and MSMC [10,11]) or by analyzing the allele frequencies of markers in a population (such as δaδi, PopSizeABC or fastsimcoal [12–14]), to recover the time to the most recent common ancestor across the genome. These methods make inferences based on the mutation clock and thus have low resolution at recent timescales [10,11]. Haplotype-based methods characterize the distribution of IBD segments in a population to infer recent demographic history [15,16]. Most commonly used IBD-based methods, e.g. DoRIS [16] and IBDNe [15], recommend the use of phased data that can be obtained from computational phasing of population data; however, this typically requires large numbers of high quality samples. Errors in computational phasing ("switch errors") can result in biased estimates of IBD segment lengths, and in turn population size inference in real data [16]. As the rate of phasing errors is inversely proportional to the sample size and the length of IBD segments [17], IBD-based methods tend to be noisy and less accurate at older timescales and when using sparse datasets with small sample sizes or limited coverage such as ancient genomes.

A third class of methods, introduced by Reich et al. (2009), characterizes the average allele sharing correlation across individuals in a population to infer the time of the founder event [18]. This approach uses the insight that a founder event introduces long-range linkage disequilibrium (LD) or allelic correlation in nearby loci co-inherited from a common ancestor by a pair of individuals in a population. As recombination occurs in each generation, it breaks down these associations over time. Thus, by measuring the decay of allelic association or LD across the genome at sites that are shared between pairs of individuals (i.e., inherited identical by state (IBS)), the time of the founder event can be inferred. A major advantage of this approach is that it does not require phased data or explicit identification of IBD segments, making it suitable for sparse datasets.

In this article, we introduce *ASCEND* (Allele Sharing Correlation for the Estimation of Non-equilibrium Demography) that extends the idea introduced in Reich et al. (2009) with two major improvements: (i) we estimate the strength of founder events in addition to their timing, and (ii) we implement the fast Fourier transform to make the approach computationally tractable, allowing us to survey large datasets. Further, we provide theoretical expectations for leveraging allele sharing correlation for estimating the parameters of the founder events. We report simulations under a range of demographic scenarios to assess the reliably of *ASCEND* and apply the method to empirical datasets from two species—humans (using present-day and ancient samples) and modern dog populations—to characterize the spatiotemporal patterns of founder events in their history.

## Results

### Overview of *ASCEND*

*ASCEND* leverages the allele sharing correlation across the genome to infer the time and strength of the founder event in a population. There are three main steps in *ASCEND*. First, for each single nucleotide polymorphism (SNP) in the genome, we infer the number of alleles that are shared IBS across pairs of individuals in the population (assuming one shared allele for heterozygous sites regardless of haplotype phase). Next, for pairs of SNPs in the target population, we compute the correlation using shared allele counts across individuals, instead of using individual genotypes (referred as *within-population allele sharing correlation*). To account for ancestral allele sharing inherited from the common ancestor, we subtract the allele

sharing inferred in pairs of individuals from the target population and an outgroup (referred as *cross-population correlation*). Finally, we measure the decay of the allele sharing correlation across SNPs separated by increasing genetic distances (Methods). This statistic is expected to decay exponentially with genetic distance and the rate of the decay is informative of the founder age (in generations) while the amplitude is related to the strength of the founder event. Intuitively, the more recent and stronger the founder event, the more correlated the shared alleles will be at short genetic distances and hence the amplitude of the exponential decay will be higher and the rate of decay slower. The strength of the founder event is captured as a composite parameter, referred to as the founder intensity ($I_f$) which is related to both the duration of the bottleneck ($D_f$) and population size ($N_f$) during the bottleneck (following [9], we define $I_f = \frac{D_f}{2N_f}$). This parameter is proportional to the probability of coalescence during the bottleneck and provides insights about the non-equilibrium response of variant frequencies to a population bottleneck [9,19].

To characterize the reliability of *ASCEND*, we simulated data under a range of demographic scenarios. First, we generated data for a single epoch bottleneck model where the target population had a founder event that occurred $T_f$ (= 10 and 300) generations ago such that the population size reduced to $N_f$ (= 5 to 500) for a short duration $D_f$ (= 1 to 30) generations. After the founder event, the population recovered to its original size $N_o$ (= 12,500) (Fig 1A). Applying *ASCEND* to the target population and accounting for cross-population allele sharing correlation with an outgroup (a distantly related group that does not share the bottleneck with the target), we found that *ASCEND* reliably inferred the age and intensity of the bottleneck when the founder event occurred within the past 200 generations (Fig 1B). We note that in addition to its high sensitivity to detect recent founder events, *ASCEND* also has a low false discovery rate and gives reliable results in the absence of founder events. We simulated a three-population demographic model to represent three modern human populations, including a West African

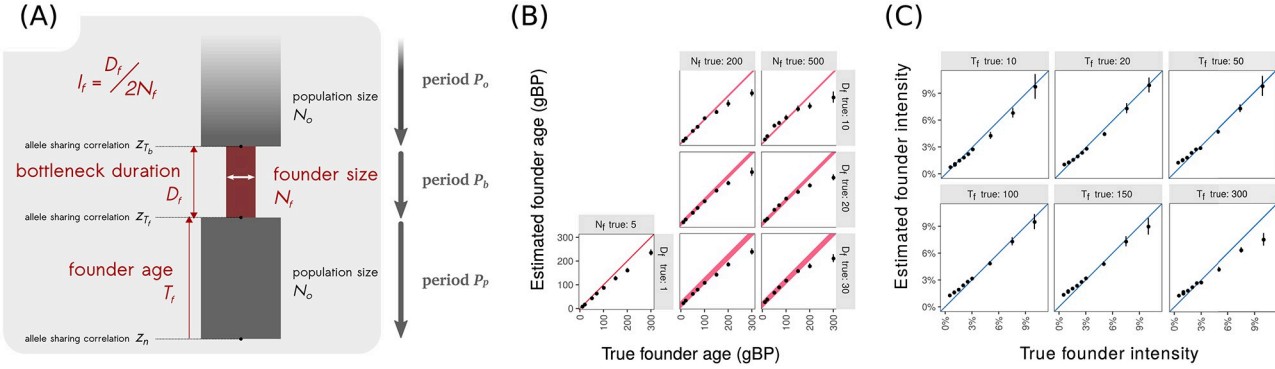

**Fig 1. *ASCEND* simulation results.** *(A) Model of founder event.* Consider a population which has experienced a founder event in its past. This history can be divided into three main periods (from the most ancient to the most recent): a period $P_o$ where the population has a constant effective population size of $N_o$, followed by a period $P_b$ where the effective population size reduces to $N_f$ for the duration of $D_f$ generations till $T_f$ generations before present. Then, the population recovers and the population size returns to $N_o$ during the period $P_P$. We simulated two populations, population *A* (target) which experienced a founder event and population *O* (outgroup, no founder event, with constant size $N_o$) that diverged 1,800 generations ago. We ran *ASCEND* and compared the estimated parameters with the true parameters of the founder event in population *A*. *(B) Accuracy in estimating founder age.* The *X*-axis shows the true founder age that was simulated in generations before present (gBP) and the *Y*-axis shows the founder age estimated by *ASCEND*. The diagonal represents the expectation (i.e., the case where the estimated values are the same as the true values). We note that for $D_f > 0$ we show a thick band for the diagonal, proportional to duration of the founder event. *(C) Accuracy in estimating founder intensity.* We define the founder intensity as the ratio of the bottleneck duration over twice the effective population size during the bottleneck, i.e. $I_f = D_f/(2N_f)$. The *X*-axis shows the true founder intensity, and the *Y*-axis shows the estimated founder intensity. The diagonal represents the expectation (i.e., the case where the estimated values are the same as the true values).

population with constant population size and two non-African groups—Northern Europeans and East Asians—with a history of out-of-Africa bottleneck and recent expansion but no recent founder events (in the past 200 generations). We found that *ASCEND* accurately failed to detect a significant founder event in all three populations using the criteria described in Methods (Notes S2.8 in S1 Text).

Next, we simulated the target population with a more complex history involving gene flow and multiple population bottlenecks. We generated data for a target population with recent gene flow that occurred $\sim 100$ generations ago that was followed by a founder event (Notes S2.4.1 in S1 Text). Applying *ASCEND* to the target population and using one of the ancestral groups as the outgroup to compute cross-population allele sharing, we inferred accurate estimates for both founder age and intensity and observed no significant impact of admixture on the inferred parameters (Fig K in S1 Text). We also simulated data for a target group with a history of two successive founder events, separated by a period of 10–200 generations (Notes S2.3 in S1 Text). We found that under this scenario, *ASCEND* reliably recovered the intensity of the strongest founder event. For severe bottlenecks ($N_f = 5$), we recovered the age of the most recent founder event, though for less severe bottlenecks, the estimated age was intermediate between the two founder ages, proportional to the weighted average of the timing of the two events (roughly weighted by intensity) (Fig I in S1 Text).

An important feature of *ASCEND* is that it does not require phased data, which makes it suitable for datasets with small sample sizes and low coverage such as ancient genomes. To test the reliability of *ASCEND* for application to sparse datasets, we investigated the impact of: (i) sample size; (ii) missing data, and (iii) features of ancient DNA samples that include (i) and (ii) along with the use of pseudo-haploid genotypes—a common practice in ancient DNA studies, where due to low sequencing coverage, the diploid genotype is determined by selecting a single random allele observed in the reads mapped at a particular site [20]. We observed that *ASCEND* estimated the parameters of the founder event accurately for target groups with greater than 5 samples (Fig R in S1 Text), even with a high rate (up to 70%) of missing data (Fig S in S1 Text). Comparing the results obtained using pseudo-haploid and diploid genotypes, we found that the estimated founder ages were similar, though the founder intensity was underestimated with pseudo-haploid genotypes (Fig V in S1 Text). This is expected because pseudo-haploid data lack heterozygous sites and thus cross-over events at short distances are missed, leading to an inflation in the variance of the allele sharing correlation. We show that by applying a correction based on the sample heterozygosity (referred as *weighted allele sharing covariance*), we obtain unbiased estimates for the founder intensity even in samples with large amount of missing data (Methods, Fig W in S1 Text). We thus use the weighted allele sharing covariance for ancient DNA and sparse present-day samples.

Finally, we implemented the fast Fourier transform (FFT) to make the allele sharing correlation calculations computationally tractable. The naïve approach for computing pairwise correlations across hundreds of thousands of markers ($n$) in the genome can be exceedingly slow in large datasets, requiring a runtime of $O(n^2)$. Following Loh et al. (2013) and considering the similarity with admixture LD calculations, we computed allele sharing correlations using FFT (Methods). Using simulations for a range of parameters, we show that the FFT approach is up to 50 times faster and provides nearly identical results to the naïve implementation (Table D in S1 Text). This allows us to apply the method to large genomic datasets.

## Founder events in present-day human populations

We applied *ASCEND* to genome-wide data from 3,102 present-day individuals genotyped on the Affymetrix Human Origins array that are part of the Allen Ancient DNA Resource

(AADR, v37.2 release) (referred as HO37 henceforth). We limited our analysis to all groups with a minimum of 5 samples. To ensure we are characterizing founder events, and not consanguinity that can also lead to long-range IBD sharing among individuals, we removed all individuals with evidence of recent relatedness (Methods). After filtering, we retained 2,310 present-day individuals from 184 groups (S1 Table and Notes S3 in S1 Text). Unless otherwise stated, we used a random set of 15 unrelated individuals to estimate the cross-population allele sharing correlation.

We first assessed the reliability of *ASCEND* in real data by comparing our results with previous publications. *ASCEND* is an extension of the allele sharing correlation statistic introduced in Reich et al. (2009) that was applied to date founder events in India. Applying *ASCEND* to this dataset, we obtained highly concordant results for all groups except one (Sahariya), where the fit in the original study looked noisy (Notes S5.1 in S1 Text). We also applied *ASCEND* to Ashkenazi Jews (AJ) and Finns that have been previously studied for their history of founder events [3,21–23]. Applying *ASCEND* to nine Finns and seven Ashkenazi Jews (AJ) in the HO37 dataset, we obtained significant evidence of founder events in both groups (S2 Table). We inferred that the founder event in Finns occurred ∼120–245 generations ago, consistent with the separation of the western and eastern areas of Finland and the arrival of the Corded Ware Complex in this region [3,24]. Similar to previous reports, we found that the founder intensity in Finns was higher than in Ashkenazi Jews [25]. In AJ, we inferred the founder event occurred 23–51 generations ago with an intensity of 0.013–0.021 (henceforth, reported as percentages for easier readability, i.e., 1.3%–2.1%) (95% confidence interval). Our estimates are consistent with a previous study that used 128 whole genome sequences of AJ and inferred a founder age of ∼25–50 generations ago and effective population size during the bottleneck of ∼250–420 that translates to an intensity of ∼1.8–3% (assuming the average bottleneck duration of 15 generations) [21,23]. This demonstrates the reliability of *ASCEND* in real data with few individuals and SNP genotypes alone.

Lastly, we confirmed that our results are reliable in groups with complex demography, particularly those involving admixture events. We compared the direct estimates of founder ages and dates of admixture inferred using genomic dating methods such as *GLOBETROTTER* and *ALDER* [26,27]. Across 64 worldwide populations, there was no significant correlation between estimated founder ages and average dates of admixture ($P$ = 0.77 for *ASCEND* and *GLOBETROTTER*; $P$ = 0.10 for *ASCEND* and *ALDER*) (S3 Table). This suggests that the inferred founder ages are not confounded by long ancestry blocks inherited through admixture, as seen in simulations (Notes S2.4 in S1 Text).

We applied *ASCEND* to study the global patterns of founder events in recent human history. We found that 61% of the analyzed populations (113 out of 184) experienced a significant recent founder event that occurred in the past 200 generations (Fig 2). The most extreme founder event (with highest intensity) was observed in the Onge population from the Andaman Islands (20.6%–21.2%), almost 10-fold higher than in AJ (Fig 2). The Onge are a demographically small and historically isolated population [28]. Demographic records suggest that this population has maintained a historically low population size and has a current *census* size of ∼100 individuals (http://censusindia.gov.in/). Across continental groups, we found that the frequency of founder events varied significantly; with the highest frequency in Oceania (80% out of 5 groups) and Americas (78% out of 9 groups) and the lowest proportion in Europe (38% out of 30 populations) (Fig 2). In addition to the frequency, the average founder intensity differed significantly across continental groups (Kruskal-Wallis test, $P$ = 7x10$^{-5}$). The founder ages ranged from ∼10 generations (in Aleuts) to 195 generations (in Icelanders) or ∼280–5,460 years, assuming 28 years per generation [29,30]. We found no correlation between

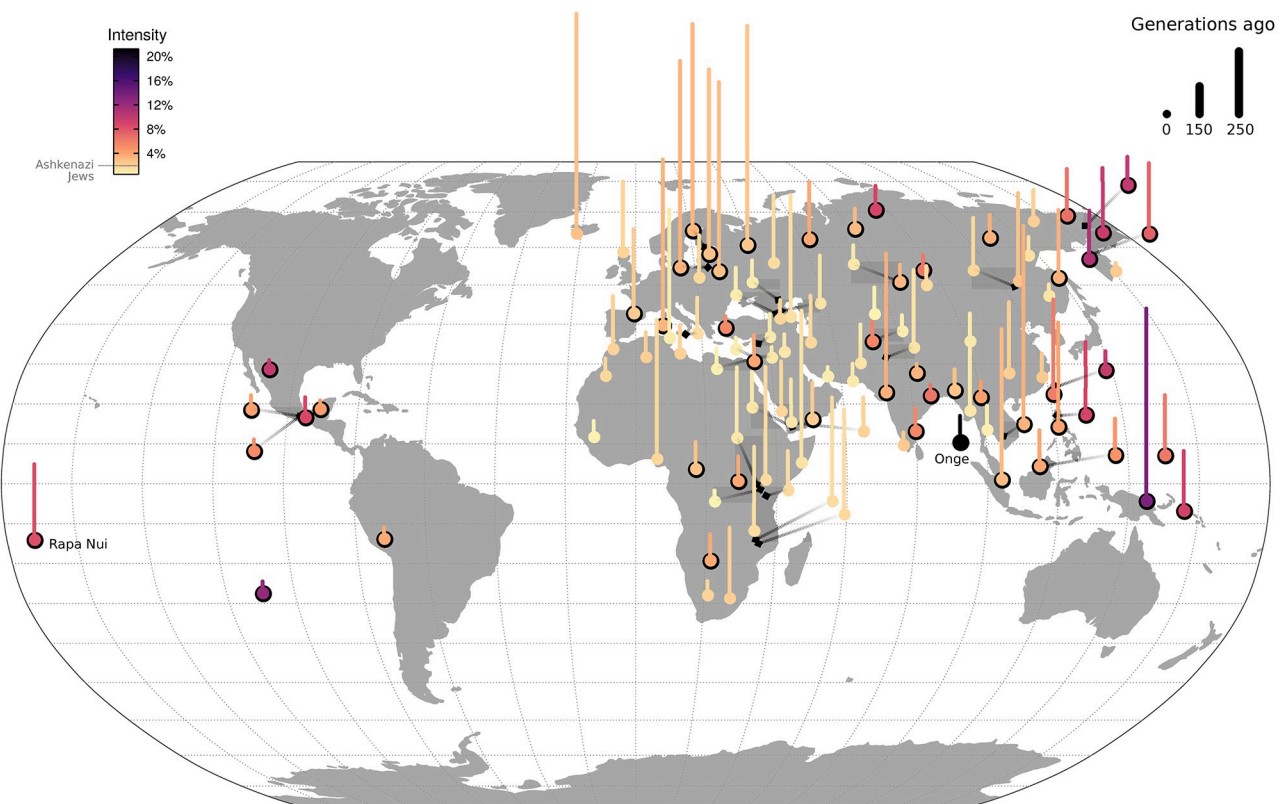

**Fig 2. History of founder events in present-day human populations.** Results of *ASCEND* for present-day populations in the Human Origins v37 dataset that passed filtering criteria and showed significant evidence of founder events (see Methods). Each point shown represents a population and the vertical segment represents the age of its associated founder event (where the segment length is proportional to the founder age). To avoid overplotting in certain areas, we shifted the location of a few populations and indicated their original location (black diamond point) with an arrow getting darker towards the original location. The color gradient of the points and segments is proportional to the estimated founder intensity. Points with a black border represent populations which have experienced significantly stronger founder events than Ashkenazi Jews (shown in legend for reference). The strongest founder event is estimated for the Andamanese population Onge (21.2%). The world map was obtained from the R package *maps* with GPL-2 public license.

founder age and founder intensity, suggesting that we can reliably disentangle the estimation of both parameters ($P = 0.99$).

Across worldwide populations, we identified 53 groups that have experienced more extreme founder events (with significantly higher founder intensity) than AJs, who have high rates of recessive diseases due to their history of founder events [1,21–23]. These populations are particularly interesting from a population and medical genetics perspective to understand the genetic consequences of population bottlenecks. Below we highlight a few notable patterns of founder events across worldwide groups and provide detailed results in S2 Table.

**Jewish communities.**    Our dataset includes samples from 11 Jewish groups, including Ashkenazi, Caucasus (Georgian), Middle Eastern (Turkish, Iranian and Iraqi), African (Moroccan, Libyan, Tunisian and Ethiopian), and Indian (Cochin) Jewish communities. We observed significant evidence of founder events in most Jewish groups, except in Ethiopian and Turkish Jews (S2 Fig). While many studies have focused on understanding the founder events in AJ, we found that the founder intensity estimated in most other Jewish groups was higher than in AJ, with the exception of Middle Eastern Jews. These results are in line with previous results based on runs of homozygosity (ROH) or IBD analysis [31,32]. The estimated

founder ages varied significantly across Jewish communities, ranging between 280–1,300 years ago, highlighting recent population bottlenecks that have impacted the genetic diversity in each group, but still older than what could be expected by consanguineous marriages with the presence of very long runs of homozygosity (ROH) [32]. The strongest and most recent founder event was inferred in Cochin Jews, with the inferred timing similar to previous reports [33].

**Islands vs. continental groups.** We observed that island populations had more extreme founder events compared to continental groups. Using data from 16 islands and 97 continental groups, we found that on average island groups had a ∼2.5-fold higher founder intensity than estimated in continental groups ($P = 3 \times 10^{-3}$, bootstrap resampling) (Fig 3A). Following Onge in South Asia, populations in Oceania ($n = 4$ populations) and Southeast Asia ($n = 6$) were found to have experienced very strong founder events. For instance, the founder intensity inferred in island groups from Papua New Guinea, Philippines and Taiwan is almost five- to ten-fold higher than in AJ. In Europe, groups from Iceland, Malta, Orkney and Sardinia had estimated founder intensities on par with AJ or more extreme, suggesting these groups had strong historical bottlenecks. In most cases, the founder ages postdate the estimates for the first settlement of the islands, suggesting ongoing population bottlenecks in many groups after their initial habitation (S2 Table).

**Hunter-gatherers, indigenous and nomadic groups.** We detected strong founder events related to nomadic lifestyle and modes of sustenance across the analyzed populations. In Africa, many hunter-gatherer groups had significant founder events including Biaka pygmies,

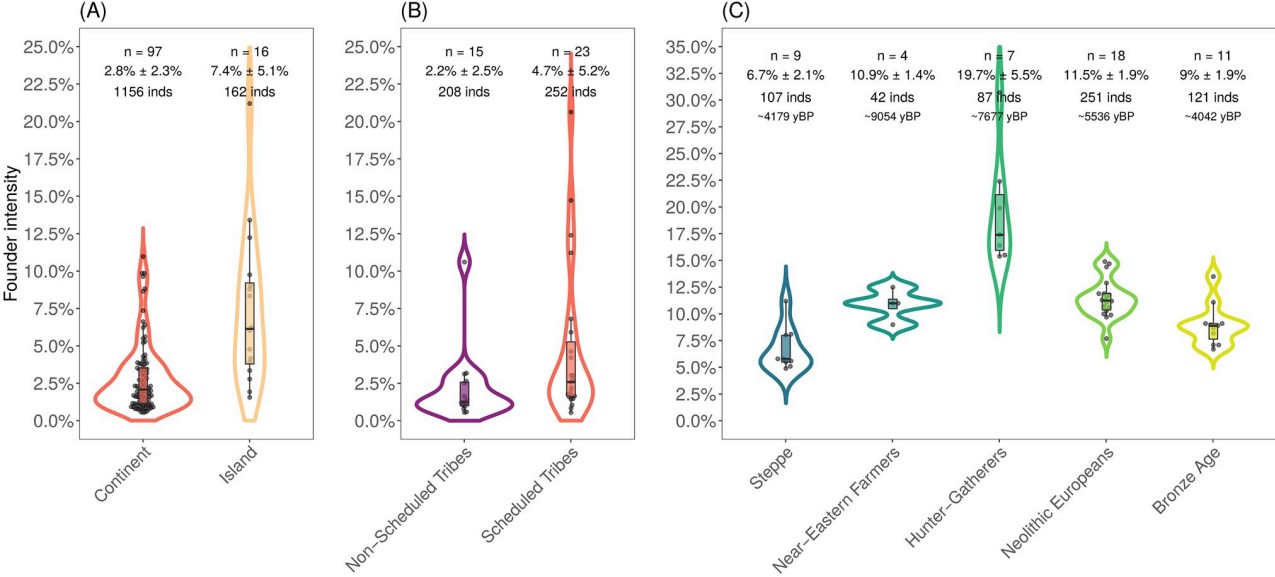

**Fig 3. Geographic and cultural practices impact founder events in humans.** We show the variation in estimated founder intensity as a violin plot across groups, classified in three plots. Each violin plot includes a boxplot and the number of populations (*n*) in each group along with the mean ± standard deviation and the total number of individuals used in the analysis. Note that within each panel, the areas of violins are the same. (*A*) *Continental vs. island populations.* This plot shows the variation in founder intensities estimated for present-day populations in the HO37 dataset classified according to geography. (*B*) *Tribal vs. non-tribal groups in South Asia.* This plot shows the variation in founder intensities estimated in the South Asian groups from the IndiaHO dataset. (*C*) *Ancient hunter-gatherers, Near Easter farmers, European Neolithic farmers, Steppe pastoralists and Bronze Age populations.* This plot shows the variation in estimated founder intensities for ancient groups in the HO44 dataset, classified based on their mode of sustenance. Below the number of individuals used in the analysis, we report the median radiocarbon age of each category in years BP (yBP). We note that in order to increase the number of groups in each category, we considered populations where the estimated founder age was dated below 300 generations before sampling age (default for other analyses was below 200 generations).

Mbuti pygmies and Ju|ʼhoan hunter-gatherers from South Africa (Fig 2). These founder events occurred recently within the past 10–20 generations. Our estimate for Mbuti pygmies (18–24 generations ago) is consistent but more precise than previous estimates (10–100 generations [34]). We also documented strong founder events in nomadic groups from Yemen Desert and Bedouins (Fig 2). Previous studies have highlighted high rates of consanguineous marriages in Bedouins [35,36]. Given that we removed recent relatives from our analysis, these results indicate that Bedouins likely experienced strong recent founder events, in addition to a history of consanguineous marriages [36]. Most of the studied indigenous Northeast Asian groups—including Aleut, Chukchi, Eskimos and Yakut—had evidence for extreme founder events (with median intensity almost three-fold higher than AJ) in the past 1,000 years (Fig 2).

**European colonization of the Americas.**   Native American groups have experienced major population declines associated with the impact of European colonization. The precise extent of this decline and the timing are, however, still debated [37]. Application of *ASCEND* to seven Native American groups from Central and South America showed evidence for significant recent founder events (S2 Table). Despite recent European gene flow in most groups [38], we inferred the median founder intensity in Native Americans was almost three-fold higher than in AJs, ranging between two- (in Quechua) to seven-fold (in Rapa Nui) higher intensity than AJ (Fig 2). We inferred the founder event occurred ∼200–500 years ago (Fig 2), postdating the European colonization of the Americas [39]. The strongest founder event was documented in Rapa Nui that occurred ∼260 years ago, coinciding with the migration of Europeans to the island [40].

**South Asia.**   Among contemporary populations, we found that the majority (64%) of South Asian groups in the HO37 dataset had more extreme founder events than in AJs (Fig 2). To investigate this history in more detail, we analyzed a larger dataset of 1,662 individuals from 249 South Asian ethno-linguistic groups genotyped on the Human Origins array [25] (referred to as IndiaHO dataset). After filtering, we found that 56% (66 out of the 118 groups) showed significant evidence of founder events (Methods, S1 and S3 Figs). Estimated founder intensities were strongly correlated with the IBD scores (a measure of the strength of founder events) calculated in an earlier study [25] (Pearson's $r = 0.95$, $P < 10^{-5}$), highlighting the reliability of using allele sharing correlations to infer the strength of the founder event (Notes S5.2 in S1 Text). In concordance with patterns seen in other worldwide regions, we observed indigenous and tribal groups ($n = 23$) had significantly stronger founder events than other groups ($n = 15$) (Kruskal-Wallis test: $P = 0.024$) (S2 Table and Fig 3B).

To understand the demographic processes leading to these extreme founder events in South Asia, we investigated the timing of the founder events across diverse ethno-linguistic groups. Previous studies have shown that most present-day Indians have ancestry from two divergent ancestral populations: Ancestral North Indians (ANI) related to Central Asians and Iranians, and Ancestral South Indians (ASI) distantly related to the Onge population [18,41]. Ancient DNA analyses have further shown that both ANI and ASI are in turn mixtures of ancient groups of South Asian hunter-gatherers, Iran Neolithic farmers, and Eurasian Steppe pastoralists [27]. Application of *ASCEND* revealed that founder ages ranged between ∼115 years ago (Gujjar) to ∼3,500 years ago (Gujaratis), with the majority of the founder events occurring within the past 1,000 years (S2 Table). Comparing the founder ages with dates of ANI-ASI admixture [27], we found that founder ages significantly postdated the admixture dates in most groups ($P = 2.2 \times 10^{-5}$) [27]. There were no significant differences in founder ages across speakers of the four major language families spoken in India (i.e., Austro-Asiatic, Dravidian, Indo-European and Tibeto-Burmese languages) (Kruskal-Wallis test: $P > 0.05$), suggesting that the spread of languages is not associated to the founder events in these groups.

## History of founder events in the ancient human past

To investigate founder events deeper into the human evolutionary past, we applied *ASCEND* to ancient DNA samples from the v44.3 release of AADR (that has a larger set of ancient genomes compared to v37.2). We limited the analysis to unrelated individuals from populations with at least 5 individuals. For ancient DNA samples, it is difficult to match the outgroup population as samples are from different timescales and geographic locations. Thus, we used the within-population allele sharing weighted covariance to avoid any bias introduced by the choice of outgroup as well as use of pseudo-haploid genotypes (Methods). We verified the reliability of this approach by simulations (Note S2.7.3 in S1 Text). After filtering, we retained 1,947 individuals from 164 worldwide ancient populations, though the vast majority of the samples were from the Americas and West Eurasia (Notes S3 in S1 Text). Our sampling ages fall in the range of ∼100–15,000 years inferred using radiocarbon dating or based on the cultural context of the specimens (S1 Table). We note that most groups older than 15,000 years had less than five samples and thus were excluded from further analysis.

Applying *ASCEND*, we discovered that 36% ($n$ = 60) of the ancient groups had significant founder events that occurred 200 generations before the individuals lived (S5 Fig and S2 Table). Overall, the median intensity across these groups was ∼9.2%, around four-fold higher than the median in present-day individuals. The strongest founder intensity was estimated in the ancient samples from Cuba—with nearly two-fold higher intensity than present-day Onge—suggesting that Cuba was settled by a small group of individuals or has maintained historically low population sizes for many generations. In general, we observed both the frequency and average founder intensity were significantly higher in groups from the Americas (frequency of 53% out of 9 analyzed groups with intensity of 10.4%–45.9%) compared to groups from West Eurasia (32% of 127 groups with intensity of 2.2%–30.7%) ($P$ = 3×10$^{-5}$). We inferred the founder events occurred ∼5–200 generations before the individuals lived. Accounting for the mean generation time and sampling age of the ancient specimens, these results translate to estimated founder ages of ∼500–17,000 years before present (BP). We noticed a trend of decreasing frequency of founder events with age—lowest frequency towards the present despite large sample sizes at recent timelines (S7 Fig, Kolmogorov–Smirnov test: $P$ = 2×10$^{-12}$), highlighting the recent human population growth.

In the ancient Americas, we observed evidence for three main founder events. Using the Lapa do Santo individuals from Brazil, dated 9,500 years BP, we estimated a founder event that occurred ∼11,900–12,700 years ago. We obtained similar dates using the Sumidouro samples from Brazil, though many samples in this group have unusually high transition-to-transversion ratio suggesting potential data quality issues [42]. These founder ages overlap estimated dates of the settlement of the Americas [43]; however, they should be viewed as tentative evidence considering the data quality issues. Another bottleneck we inferred occurred ∼5,500–6,000 years BP in ancient samples from three Caribbean islands (Cuba, Dominican Republic and Bahamas) with diverse sampling ages, ranging between 470 to 2,300 years. Finally, two ancient populations from the Pacific coast of North America (San Nicolas Island from California and Aleut from Alaska) showed evidence for founder events between 2,400 and 2,800 years BP. Unlike in present-day samples, these dates overlap archeological evidence for the peopling of the various islands as the samples are closer to the timing of first settlements in these islands. Finally, we found ancient groups from the Americas had markedly stronger founder events than present-day individuals from this region, with on average four-fold higher intensity.

Recent analysis has shown that present-day Europeans are a mixture of three major ancestry groups related to ancient European hunter-gatherers, Anatolian farmers, and Eurasian Steppe pastoralists [44]. Previous comparisons of diversity patterns in ancient samples suggested that ancient hunter-gatherers had very low genetic diversity [45]. However, these inferences were based on low coverage data, where calling diploid genotypes is challenging and hence estimates of heterozygosity can be uncertain [45,46]. We applied *ASCEND* to ancient west Eurasian samples and compared the intensity of founder events (a proxy for genetic diversity in the population) across groups. Specifically, we used European hunter-gatherers (sampled between 6,394–9,721 years BP), Near Eastern farmers (4,749–9,958 years BP) and Eurasian Steppe pastoralists (3,505–7,530 years BP). To expand our sample size, we considered founder events that occurred in the past 300 generations before the individuals lived (note, this is older than the threshold of 200 generations used elsewhere, but we carefully inspected that the fitted curves were reliable). Across the three groups, we found that the frequency of founder events was similar, ranging between 90–100%. However, the average founder intensity was significantly higher in European hunter-gatherers (15.4%–29.5%, interpercentile range at 95%) compared to the Near Easter farmers (9.1%–12.4%; $P = 0.0053$) or the Steppe pastoralists (4.9%–10.6%; $P < 10^{-20}$) (Fig 3C). Similar results were obtained when we restricted the comparison to groups with founder ages below 200 generations ($P < 0.01$ for both pairwise comparisons). This highlights the impact of modes of sustenance (foraging, farming and pastoralism) on human population sizes, mirroring the pattern seen in the analysis of present-day individuals.

Turning to time transects across Europe, we compared hunter-gatherers, Neolithic farmers and Bronze Age individuals across diverse regions. These samples are associated with major changes in lifestyle, technologies and genetic ancestry across Europe, coupled with high rates of population growth. Comparing the founder events across ancient Europeans, we estimated that the frequency of founder events decreased over time, varying between 100% in hunter-gatherer to 78% in Neolithic Europeans and finally, 55% in the Bronze Age individuals. Further, we found that the intensity of founder events decreased significantly across these three periods of transition (Kruskal-Wallis test, $P = 1 \times 10^{-5}$) (Fig 3C). Interestingly, we note that founder ages in hunter-gatherers, Neolithic and Bronze Age populations have overlapping intervals with the medians across groups ranging between 9,000–15,000 years ($P > 0.05$). Moreover, founder intensity in these populations was strongly and positively correlated with their sampling age ($P < 10^{-7}$). These results suggest that the founder events could be in part or fully related to a shared founding bottleneck in the ancestors of Europeans and the founder intensity has decreased over time, possibly due to gene flow from Near Eastern farmers and Steppe pastoralists.

We estimated the oldest founder event in Upper Paleolithic individuals sampled from the Taforalt Cave cemetery (eastern Morocco). These individuals were associated with the Ibero-maurusian culture of microlithic bladelet technology from North Africa [47]. A recent study found that these individuals can be modeled as a mixture of Near Eastern hunter-gatherers (Natufians) and sub-Saharan Africans, with no apparent gene flow from the Epigravettian culture of Paleolithic southern Europe [47]. Due to the staged appearance of microlithic bladelet technologies and its rare geometric form, it had been suggested that population structure, population bottlenecks or intermittent isolation of populations in North Africa could potentially explain the lack of continuity in stone tool cultures in this region [48]. Indeed, we found evidence for a significant founder event occurring $\sim 16,700$ years BP with a founder intensity of $\sim 20\%$ [16.3%–23.7%] similar to the strongest founder event inferred in present-day human populations (S3 and S5 Figs). Our results support the hypothesis that a relatively small group of individuals developed the Iberomaurusian tool culture.

### History of recent and extreme founder events in canids

To demonstrate the general applicability of our method, we applied *ASCEND* to data from modern dog breeds. The domestication and establishment of various dog breeds were accompanied by severe founder events and selection [49]. To reconstruct the history of strength and timing of these founder events, we used data from ∼6,000 domesticated dogs (∼200 breeds, including populations of village dogs and mixed breeds) from two publicly available datasets from the Sams [50] and Hayward [51] studies. We excluded individuals with evidence of recent inbreeding or close relatedness and considered only groups with more than 5 individuals. After filtering, we retained 52 populations belonging to 40 unique breeds and two village dog populations (Methods, S4 Table, Notes S6 in S1 Text).

Application of *ASCEND* to canids revealed significant evidence of founder events in all populations analyzed, with an average intensity of ∼25.3% across breeds (ranging between 1.3% in village dogs to 77.7% in Boxers) (Fig 4 and S5 Table). Note, a subset of SNPs used in this analysis were ascertained in Boxers and the canid reference genome (CamFam) also derives from a Boxer individual. These factors could lead to increased power to detect founder events in Boxers, though it should have minimal impact on other breeds [52–54]. Interestingly, founder intensities differed significantly across the traditional roles of dog breeds, with significantly higher estimates in traditionally agricultural or sedentary breeds (non-sporting dogs or working dogs) than breeds used for hunting or sports (hounds or sporting dogs) ($P = 6\times10^{-4}$, Kruskal-Wallis test) (Fig 4). We obtained highly correlated results in the Sams and Hayward datasets (founder age: Pearson's r = 0.9, $P = 4\times10^{-4}$, and founder intensity: r = 0.92, $P = 1.9\times10^{-4}$) (Fig AD in S1 Text). Founder events in all breeds occurred very recently, within the past 25 generations (Fig 4 and S5 Table). The most recent founder event we inferred occurred ∼6 generations ago (in Gordon Setter) and the oldest ∼24 generations ago (in Bulldogs) (Fig 4). Assuming a generation time of 3–5 years [55,56], this translates to ∼75–125 years ago.

## Discussion

We introduce *ASCEND*, a two-locus approach that leverages the allele sharing correlation across the genome to infer the time and strength of the founder event in a population. *ASCEND* is complementary to IBD-based methods—such as DoRIS and IBDNe [15,16]—in its time range but is more flexible as it does not require phased data and hence is suitable for sparse datasets. By applying *ASCEND* to around 300 present-day and 160 ancient human populations, we document that more than half the groups in our study experienced a strong founder event during the past 10,000 years. To our knowledge, this is the first comprehensive survey of founder events across worldwide human populations and provides insights about the frequency and demographic processes underlying population bottlenecks during human evolution. We note that the sampling of human populations in our study is not random as the Human Origins and the Allen's Ancient DNA Resource datasets are enriched for small, isolated groups and some regions are better represented than others. Thus, the reported frequencies could differ in future worldwide surveys of human populations.

We recover previously reported signals of founder events in AJs, Finns and South Asians, as well as provide new insights and details about events in many groups such as Oceanians, Native Americans and Northeast Asians. Our results suggest that geographic isolation, modes of sustenance and cultural practices are notable predictors of founder intensity. Specifically, we document that populations living on islands have experienced stronger founder events than continental groups. This could be because island populations are formed by small numbers of individuals or because they have maintained a small population size due to limited

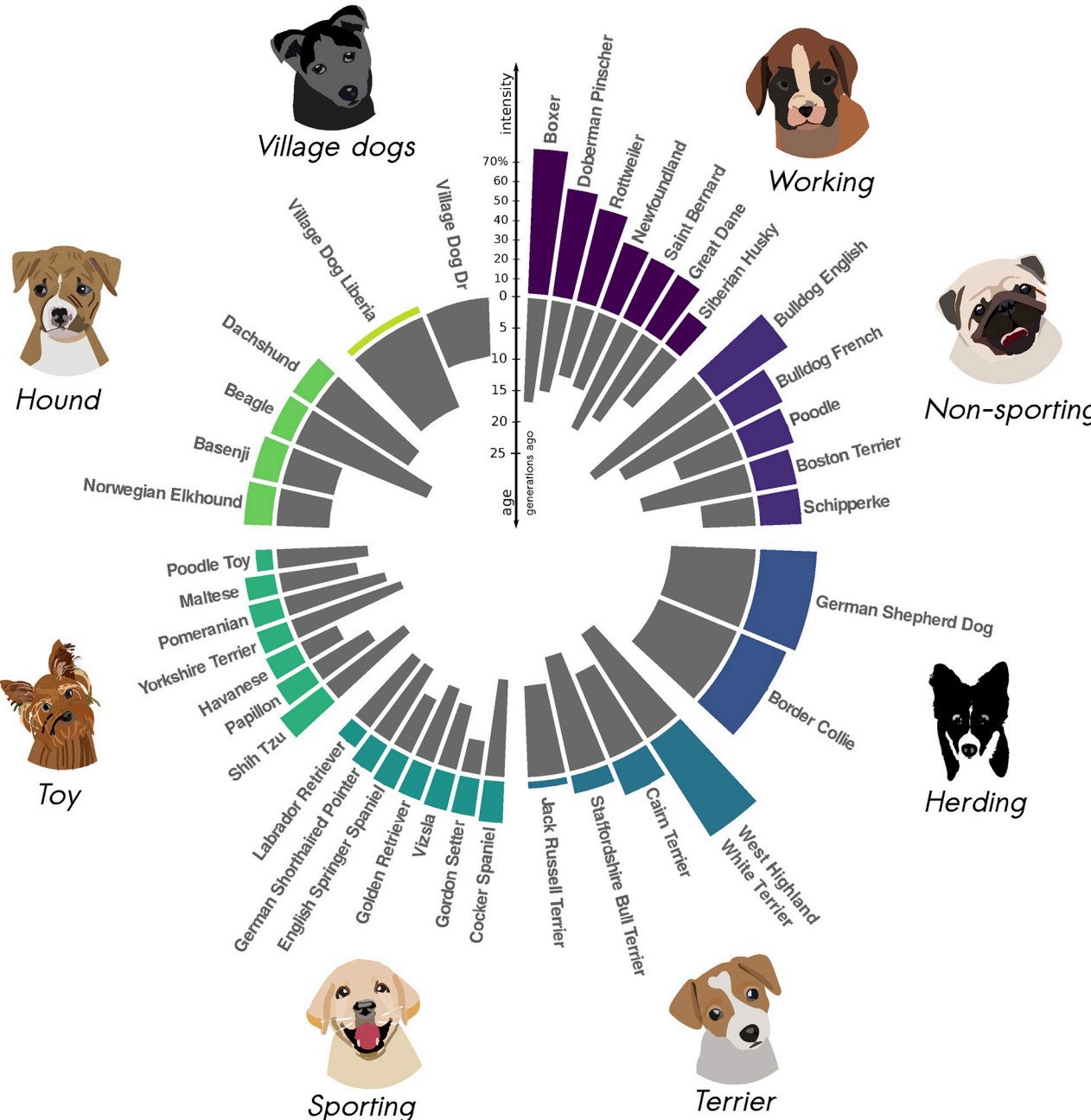

**Fig 4. History of extreme founder events in dogs.** Results of *ASCEND* for all dog breeds that passed the filtering criteria and showed evidence for significant founder events in the Hayward dataset (see Methods). The breeds are grouped according to their role as reported by the American Kennel Club database. The outer (colored) rim represents the founder intensity, with bar height proportional to the founder intensity. Note that within each role category, breeds are classified by decreasing founder intensity in the clockwise direction. The inner (gray) rim represents the founder age (in generations before present), with the bar height proportional to the founder age. The width of the bars is inversely proportional to the number of breeds within each role category. Icons were retrieved from openclipart.org.

resources. Most hunter-gatherers, nomadic and indigenous groups in our dataset had strong founder events, possibly linked to limitations of resources and extreme environmental pressures. Across diverse present-day populations including South Asia and the Americas, we document significant founder events that postdate periods of historical migrations and admixture.

Finally, cultural practices such as endogamous marriages contributed to founder events as seen in Ashkenazi Jews and South Asians.

Applying *ASCEND* to ancient human genomes, we surveyed the founder events in deep human history and pre-history. We inferred that founder intensities in ancient groups were markedly higher than in present-day individuals. As observed in present-day samples, modes of sustenance (foraging, agriculture, and pastoralism) were associated with the strength of founder events. We found ancient hunter-gatherers had stronger founder events than the Near Eastern farmers and Eurasian Steppe pastoralists. Moreover, using time transect samples from Europe, we found local hunter-gatherer groups had more extreme founder events than the Neolithic farmers or Bronze Age individuals. This suggests that population sizes in Europe have increased over time, coupled with changes in ancestry and transitions in lifestyle. Our results are consistent with a recent study that measured short runs of homozygosity in ancient Europeans and found a similar increase in population size during the Neolithic period [57]. Our results are also in agreement with archeological evidence for increased population size during the Neolithic transition [58]. This underscores the power of using simple, flexible statistics to make inferences with the limited data available from ancient DNA samples.

Our estimates of founder ages help to shed light on the historical events that led to population isolation and bottlenecks. In the Americas, we detected four main episodes of founder events which can be related to important historical events in this region. First, using ancient samples, we obtained tentative evidence for a strong founder event that occurred $\sim$12,000 years BP, concordant with the early settlement of the Americas [43]. Future ancient DNA studies with more samples will provide more precise and reliable estimates of the founding bottleneck in the Americas. The second episode was inferred $\sim$5,500–6,000 years ago using data from Caribbean regions, in agreement with archeological dates for the peopling of the Caribbean [59]. A third founder event that occurred $\sim$2,500 years ago in individuals from Aleutian Islands and San Nicolas Islands could be related to the coastal settlement of North America. Finally, we inferred recent founder events that occurred $\sim$200–500 years ago in present-day Native Americans that postdate the European colonization of the Americas [60]. Together, these results help to reconstruct the key founder events that have shaped the genetic variation in the Americas.

The large and comprehensive set of samples from India in our study—including samples from most geographical regions, speakers of all major language families and tribal and caste groups—highlights the widespread history of founder events in this region and provides insights about the origin of endogamy in India (S2 Table). In many Indian communities, marriages across caste (*varṇa*) and sub-castes (*jāti*) are restricted. Earlier writings describe the caste system—comprising Brahmana, Kshatriya, Vaishya, and Shudra—as a class structure based on occupation. The later writings especially the law code of Manu (*Manusmṛiti*) introduced restrictions against intermarriage across castes [61]; though the chronology of *Manusmṛiti* remains debated [62]. Alternatively, the origin of endogamy has been proposed to be very recent—tracing back to restrictions against intermarriages that occurred in the past few hundred years during the British Raj [63]. Our direct estimates of founder ages provide an independent line of evidence to understand the origin of endogamy in India. We inferred that these founder events occurred between $\sim$120–3,500 years ago across 78 ethno-linguistic groups in India. Our dates are consistent with a previous smaller survey including 13 ethno-linguistic groups from India [18]. In a majority of the populations, the founder events occurred within the past 600–1,000 years, suggesting this period was integral to shaping endogamy in India. These estimates pre-date the British colonization of India but postdate the ANI-ASI admixture (or spread of Iranian farmer or Steppe pastoralist ancestry to the subcontinent) [27,41]. Endogamy likely became stronger during the British Raj which could have further

contributed to the founder events in many groups. In this scenario, our dates would reflect average estimates of multiple founder events, though the patterns we observe cannot be fully explained by recent events alone.

The oldest founder event ($\sim$16,700 years BP) was dated in the ancient Taforalt individuals from Morocco, who had a ten-fold more extreme population bottleneck than present-day AJs. This group has been associated with the Iberomaurusian microlithic bladelet technology and a question has been how large was the population of the Taforalt that introduced this technology. Using the direct estimate of founder intensity in Taforalt individuals and assuming the founder duration of 15 generations (as AJ [23]), we infer an approximate effective population size of $\sim$40. Analysis of modern human populations suggests that the effective population size tends to be around one-third of the census size [64]. This translates to an estimate for the census population size of the Taforalt at roughly $\sim$120 individuals. This is similar in scale to the census size of Andamanese islanders who have a similar founder intensity as the Taforalt individuals (S2 Table). We note that these calculations make a number of simplifications and the estimated population sizes may vary depending on the demographic model (e.g., duration of the bottleneck, migration etc.). However, they provide qualitative evidence that the Taforalt individuals were descendants of a relatively small, isolated group that was on the order of hundreds of individuals.

We show the wide applicability of our approach to non-human species by applying *ASCEND* to the domesticated dog species. Among the 42 unique populations that we analyzed, all breed dogs and two village dog populations had significant evidence for a recent founder event. These founder events may be due to: (i) *inbreeding* that involves mating between closely related individuals, or (ii) *the sire effect* whereby only a very few highly valued individuals (based on selected phenotypic traits) are bred repeatedly and contribute disproportionately to the next generations [65]. Such strong founder events can lead to high rates of homozygosity, and in turn increased risk of diseases. In accordance, some breeds like Boxers and Bulldogs that have among the strongest founder events in our analysis are also known to be affected by high rates of recessive diseases. For instance, arrhythmogenic right ventricular cardiomyopathy, a cardiac disease-causing sudden death in dogs, is seen at high frequency in both breeds [66,67]. We found the temporal distribution of founder events in dogs ($\sim$75–125 years ago) overlaps with the Victorian era when a large number of modern dog breeds were created in Great Britain in the context of popularizing dog-fancying and showing [68]. Our results are consistent with a recent study that measured IBD and ROH patterns in $\sim$4,000 breed dogs and found severe bottlenecks in most breeds [69].

A caveat to our results is that we have estimated parameters of the founder events assuming an epoch model, but in fact we have not distinguished between the patterns expected under other models such as a gradual exponential growth or no recovery bottleneck model. In Notes S2 in S1 Text, we report simulations showing results for these scenarios. In the case where the population maintains a small size from the time of the bottleneck to the present (no recovery model), we were only able to recover the founder intensity and not the timing of the start of the bottleneck. For a gradual exponential model, we do not recover the founder parameters reliably. In *ASCEND*, we fit an exponential model with two parameters, age and intensity. This means that parameters like the rate of population size increase after the founder event are not captured (but assumed to be large) and thus the results should be interpreted with caution if there is evidence supporting the exponential growth model (Notes S2.5 in S1 Text). We note that non-parametric methods that characterize the distribution of IBD segment lengths can also provide biased estimates of historical effective population sizes despite modeling the exponential growth. This is because IBD segments inherited from common ancestors living during periods of high population sizes will be rare and short, thus hard to detect [70]. Further, these

methods are not applicable to groups with few individuals or low coverage ancient genomes that cannot be reliably phased. Indeed, when we applied IBDNe to the IndiaHO dataset with small sample sizes, we obtained very unstable and noisy results (Notes S5.3 and Fig AC in S1 Text).

Finally, we note that complex demographic scenarios in which admixture events postdate founder events are hard to interpret, as the founder event could have occurred in the target or one of the ancestral populations. In this scenario, the target population has a mosaic genome with chromosomal segments from multiple ancestral groups and the signatures of founder event may be present only in a subset of their genomic regions (if the founder event occurred in one of the ancestral populations). We explored this scenario by generating simulated data where the founder event(s) occurred in one or both ancestral populations of an admixed target population (Notes S2.4.2 in S1 Text). As expected, we found the intensity was lower than at the start of the bottleneck as admixture increases diversity. The inferred founder age could be biased depending on the number, the source or ancestral group experiencing the founder event, and the proportion of admixture. For scenarios with very low admixture, we were able to recover the founder age accurately. However, for higher proportions of admixture, the founder age was underestimated or similar to the time of admixture (Fig M in S1 Text). We note that in practice this bias should not impact the empirical results reported above as we observed minimal correlation between the inferred time of admixture (*GLOBETROTTER* or *ALDER*) and the founder ages (S3 Table). However, when there is evidence of recent admixture in the target population which postdates the founder event, it is advisable to perform local ancestry inference in the admixed population and then apply *ASCEND* to genomic regions that are confidently assigned to each ancestral population separately. This will lead to less ambiguity about the source of the founder event (target or one of the ancestral groups) and provide more reliable results.

In summary, we document founder events across space and time in two species and short-list groups that have experienced significant founder events in their recent history. These results imply that many present-day human populations could have an increased risk of recessive diseases, as previously documented in Finns and AJs [3,71,72]. Future disease mapping efforts should prioritize founder populations as they offer immense potential for biological discovery and reducing disease burden through the discovery and testing of recessive disease-associated genes and pathways.

## Methods

### *ASCEND*: model and theory

*ASCEND* measures correlation in allele sharing between pairs of individuals across the genome to recover the age and intensity of a founder event. Below we describe the theoretical and implementation details of *ASCEND*.

### Basic model and notation

Assume we have a target population *A* that has a history of recent founder event that occurred $T_f$ generations ago where the population size reduced from $N_o$ to $N_f$ ($N_f \ll N_o$) (Fig 1A). To estimate the properties of the founder event in *A*, we compute the average correlation in allele sharing $z_w(d)$ across pairs of individuals in *A* (referred to as *within-population allele sharing correlation*). Specifically, for each SNP *i*, we record if pairs of individuals (*a*, *b*) share 0, 1 or 2 allele (s) (i.e., number of IBS alleles), a quantity we denote as $N_{a,b,i}$. For heterozygous sites, we assume the individuals share one allele, regardless of haplotype phase. We then average the correlation coefficient *r* across all pairs of neighboring SNPs (*i*, *j*) located at a genetic distance

of $d$ Morgans apart as follows:

$$z_w(d) = \frac{1}{S_d} \sum_{S_d} r\left(\left[N_{a,b,i}\right], \left[N_{a,b,j}\right]\right) \tag{1}$$

where $S_d$ is the set of unique pairs of SNPs located $d$ Morgans apart and $|S_d|$ is the number of pairs of SNPs in the set $S_d$. $[N_{a,b,i}]$ is the vector of allele sharing values across all possible pairs of individuals $a$ and $b$ ($a \neq b$) in population $A$.

To minimize the impact of ancestral allele sharing, which may lead to allelic correlation at short distances even in the absence of a founder event, we compute a *cross-population allele sharing correlation $z_c(d)$*, which measures the correlation across pairs of individuals in $A$ and an outgroup population ($O$) (Fig 1A) as follows:

$$z_c(d) = \frac{1}{S_d} \sum_{S_d} r\left(\left[N_{a,x,i}\right], \left[N_{a,x,j}\right]\right) \tag{2}$$

where $a$ is the index of any individual from the target $A$ and $x$ the index of any individual from the outgroup $O$. Ideally, the outgroup population is a population which has separated from the target recently but does not share the bottleneck with individuals in the target group. In simulations and empirical analyses, we tried different setups to choose the outgroup individuals: (i) a set of random individuals (excluding the target population), (ii) a set of individuals from another population with similar ancestry profile as the target group (Notes S5.1 in S1 Text), or (iii) a distantly diverged population from a target. We observed that the estimated founder parameters showed little sensitivity to the choice of the outgroup.

We define the corrected allele sharing correlation $z(d)$ for any genetic distance $d$ as:

$$z(d) = z_w(d) - z_c(d) \tag{3}$$

## Relating allele sharing correlation to founder event parameters

Assume if individuals reproduce at random in a population without selection or migration, the correlation across neighboring SNPs $z$ is expected to decay due to recombination over successive generations [73]. Specifically, the amount of allele sharing correlation between two SNPs is expected to decrease due to (i) cross-over events that break down the allelic associations between SNPs, occurring as a function of the genetic distance $d$ between neighboring SNPs and (ii) due to the loss of allelic variation (i.e., heterozygosity) because of genetic drift in finite population size, occurring at a rate $\left(1 - \frac{1}{2N_o}\right)$. Thus, $z_n$, the correlation in allele sharing (i.e., linkage disequilibrium at IBS sites) at generation $n$, can be expressed recursively as a function of the correlation in the previous generation $z_{n-1}$, the genetic distance between the SNPs $d$ Morgans and the effective population size $N_o$.

$$z_n = z_{n-1}(1-d)^2 \left(1 - \frac{1}{2N_o}\right) \tag{4}$$

where $(1-d)^2$ is the probability that no recombination occurred between the two SNPs located $d$ Morgans apart in the past generation and $\left(1 - \frac{1}{2N_o}\right)$ is the probability that the alleles at the SNP $j$ do not coalesce in the previous generation. For all analyses, we assume a single recombination event occurred between two neighboring SNPs. While in practice, more than one event could occur, we consider $d$ to be small enough that the probability of multiple events is exceedingly low.

We first consider the period $P_P$ from the end of the bottleneck ($T_f$ generations ago) to present (Fig 1A). During the bottleneck, the population size reduces to $N_f$ and then recovers back to $N_o$ after the bottleneck. Thus, the allele sharing correlation can be expressed by recurrence as:

$$z_n = z_{T_f} \cdot \left( (1-d)^2 \right)^{T_f} \cdot \left( 1 - \frac{1}{2N_o} \right)^{T_f} \tag{5}$$

where $z_{T_f}$ is the allele sharing correlation at the end of the bottleneck, $T_f$ generations ago. For large $N_o$ and small genetic distances $d$, we can approximate this equation by:

$$z_n = z_{T_f} \cdot e^{-2 \cdot d \cdot T_f} \cdot e^{-\frac{T_f}{2N_o}} \tag{6}$$

Assuming $N_o \gg T_f$, $\frac{T_f}{2N_o}$ tends towards 0, so we can write:

$$z_n = z_{T_f} \cdot e^{-2 \cdot d \cdot T_f} \tag{7}$$

As we can see in Eq (7), the allele sharing correlation decays exponentially with a rate proportional to the age of the founder event [18].

Now, considering the period of the bottleneck $P_b$ (Fig 1A) for a duration of $D_f$ ($D_f < 2N_f$), we can express $z_{T_f}$ as a function of $z_{T_b}$, the correlation before the onset of the bottleneck $T_b$ (= $T_f + D_f$) generations ago, assuming no recombination events occurred during this period:

$$z_{T_f} = z_{T_b} \cdot \sum_{i=1}^{D_f} \frac{1}{2N_f} = z_{T_b} \cdot \frac{D_f}{2N_f} \tag{8}$$

where $z_{T_b}$ is the correlation between two adjacent SNPs before the onset of the bottleneck and $\frac{D_f}{2N_f}$ is the probability that the two alleles share a common ancestor during the bottleneck period $P_b$. Following Balick et al. (2015), we define the intensity of the bottleneck, $I_f = \frac{D_f}{2N_f}$. Thus, the equation above can be written as a function of the founder intensity as:

$$z_{T_f} = z_{T_b} \cdot I_f \tag{9}$$

Now consider the period $P_o$ before the bottleneck (Fig 1A). Our model assumes that the demography is at equilibrium before the onset of the bottleneck with the population size approximately equal to $N_o$. Thus, $z_{T_b}$ equals to the expected time to coalescence for a pair of lineages and is on average $2N_o$ generations (if and only if $N_f < N_o$) [74]:

$$z_{T_b} = \left( 1 - \frac{1}{2N_o} \right)^{2N_o} = e^{\frac{-2N_o}{2N_o}} = e^{-1} \tag{10}$$

Integrating the information from the above equations, we obtain the allele sharing correlation computed at the sampling time ($z_n$) as a function of founder age $T_f$ and founder intensity $I_f$ as follows:

$$z_n = I_f \cdot e^{-\left( 1 + 2 \cdot d \cdot T_f \right)} \tag{11}$$

## Use of pseudo-haploid genotype data for inference

In ancient DNA applications, due to the limited sequencing coverage, it is common to use pseudo-haploid genotypes—determined by randomly selecting one allele at each SNP and assigning it as a homozygous genotype—for the analysis. The depletion of heterozygous sites could lead to an underestimate of the frequency of cross-overs at short distances. In turn, this could lead to an inflation of the standard deviation of the allele sharing values, though the mean should remain unbiased. Consequently, the estimated founder ages remain reliable, but the founder intensity can be underestimated. The bias is proportional to the expected heterozygosity of the SNP pairs used for the analysis.

To account for this effect, we propose a weighted allele sharing covariance statistic defined as follows:

$$w\left(A_i, A_j\right) = \frac{cov\left(A_i, A_j\right)}{\sqrt{H_{e,i}}\sqrt{H_{e,j}}} \tag{12}$$

with $A_i$ the vector of allele sharing values at SNP $i$ and $H_{e,i}$ the expected heterozygosity of this SNP estimated as $2p_i(1-p_i)$ with $p_i$ the frequency of the reference allele. In simulations and real data, we show that the weighted covariance statistic provides unbiased estimates for the founder intensity even for datasets with large proportions of missing genotypes (Notes S2.7.3 in S1 Text).

## Implementation of ASCEND

We implemented the allele sharing correlation using two approaches, a Naïve approach and another approach using a fast Fourier transform (FFT).

**Naïve approach.** Using genotype data with $M$ individuals and $N$ SNPs, we construct an allele sharing matrix whose elements contain the counts $\left(c_i^\pi\right)$ of alleles shared for any $\pi^{\text{th}}$ pair of individuals at any SNP $i$. This matrix has dimensions $N \times P$, where $N$ is the number of SNPs and $P$ is the number of individual pairs: $P = \frac{M(M-1)}{2}$. We then compute the correlation across pairs of SNPs using the following equation:

$$z(d) = \frac{1}{S_d} \sum_{S_d} r\left(\left[c_i^\alpha, c_i^\beta, \cdots\right], \left[c_j^\alpha, c_j^\beta, \cdots\right]\right) \tag{13}$$

where $z(d)$ is the allele sharing correlation for a genetic distance $d$ Morgans, $S_d$ is the set of pairs of SNPs $(i, j)$ located $d$ Morgans apart, $|S_d|$ is the size of this set, $c_i^\alpha$ is the allele sharing where $\alpha$ is the index of the pair of individuals and $i$ is the index of the SNP. We compute $r$ which is the Pearson's correlation coefficient. This approach has a runtime of $O(n^2)$.

**FFT-based approach.** We describe details of the FFT implementation in Notes S1 in S1 Text. Briefly, in the FFT implementation, instead of $z(d)$, we compute the autocorrelation in allele sharing for each pair of individuals along "mesh points". This means that instead of iterating the calculation of the correlation between two vectors of allele sharing over all pairs of SNPs (Naïve approach, the number of iterations is therefore the number of SNP pairs), we now iterate the calculation of the autocorrelation along mesh points of one vector of allele sharing over all pairs of individuals (FFT approach, the number of iterations is the number of individual pairs). Since the number of individual pairs is drastically lower than the number of SNP pairs, we can see how the FFT approach offers a significant speed-up. Mesh points are equally spaced genetic positions located every $\delta$ Morgans along the genome (such that $\delta \ll d$).

Specifically, the FFT approach computes the following quantity:

$$z(d) = \frac{1}{m\binom{n}{2}}\sum_{\tau=1}^{m}\sum_{\pi=1}^{\binom{n}{2}} r_{A_\pi A_\pi}(\tau) \tag{14}$$

where $n$ is the number of individuals, $m$ is the number of mesh points in the bin $d$, $A_\pi$ is the vector of allele sharing in the individual pair $\pi$ across all mesh points (the length of $A_\pi$ is thus equal to the number of mesh points), and $r_{A_\pi A_\pi}(\tau)$ is the autocorrelation coefficient for genetic distance sub-bins (non-overlapping shifts) of $\tau$ Morgans within the distance bin $d$ (Notes S1.2 in S1 Text). This approach has a runtime of $O(n\,log\,n)$.

We note that the use of the mesh is the only source of approximation in the FFT implementation compared to the Naïve approach. The genetic distance between mesh points, $\delta$, depends on the dataset size and leads to a trade-off between runtime and accuracy. To improve accuracy, $\delta$ should be smaller than the bin size $d$ used in the Naïve implementation. Empirically, we observe that setting the mesh size as 0.001 cM gives nearly identical allele sharing correlation values between the Naïve and FFT implementations (Notes S2.9 in S1 Text). With this mesh size, we obtain a runtime speed-up of a hundred-fold, allowing the analysis of population genome-wide data to be completed in a few minutes rather than taking hours with the Naïve approach.

## Inference of parameters of the founder event

To estimate the age and intensity of a founder event in a target population $A$, *ASCEND* computes the allele sharing correlation $z(d)$ as the difference between the within-population allele sharing $z_w(d)$ across pairs of individuals in $A$ and the cross-population correlation $z_c(d)$ across the individuals in the target and outgroup populations (Eq (3)). For present-day samples, we used a random set of 15 non-overlapping individuals (excluding the target population) from the curated dataset (S1 Table). For ancient DNA samples, we did not use an outgroup as it is difficult to match samples based on the time of sampling and geographical location. Further, for all populations with pseudo-haploid genotypes—including eight present-day populations from HO37 (annotated with the suffix ".*SG*") and all the ancient populations from HO44—*ASCEND* was run using the weighted allele sharing covariance (S1 Table). We ran *ASCEND* for sites having at least one variant present in at least one sample across the target and outgroup populations, and for pairs of SNPs with at least one non-missing genotype in common across the individuals. Unless otherwise noted, we computed $z(d)$ for genetic distances ranging from 0.1 to 30 cM, with a bin size of 0.1 cM. For the FFT implementation, we set the number of mesh points per bin to 100. For dogs, we set the maximum genetic distance to 40 cM (instead of 30 cM) as we observed long-range LD extending to large distances in many breeds.

To estimate the age and intensity of the founder event, we fitted an exponential distribution using non-linear least squares to estimate the rate and amplitude of $o(d)$ measured as $z(d)$ (Eq (3)) using diploid genotypes or $w(d)$ using pseudo-haploid genotypes (Eq (12)), respectively:

$$o(d) = a \cdot e^{-2 \cdot d \cdot t} + c \tag{15}$$

Here, the affine term, $c$, is used to account for noise in the fit of the exponential decay, which is a function of the sampling variance. We then simultaneously estimate the parameters of interest: $\hat{T}_f$, the age of the founder event and $\hat{I}_f$, the intensity of the founder event as follows:

$$\hat{I}_f = e^1 \cdot a \tag{16}$$

and

$$\hat{T}_f = t \tag{17}$$

Note that if the genetic distances are given in cM:

$$\hat{T}_f = 100 \cdot t$$

We compute standard errors by performing a weighted block-jackknife procedure [75], considering each chromosome as an independent block and setting the weights proportional to their respective SNP counts. For all analyses, we report the 95% confidence interval of the estimated parameters.

In order to assess the exponential fit, we calculate a normalized root-mean-square deviation (NRMSD) between the empirical allele sharing correlation values $z$ and the fitted ones $\hat{z}$, across the all the genetic distance bins (noting $D$ the number of bins):

$$NRMSD = \frac{1}{max(\hat{z}) - min(\hat{z})} \sqrt{\frac{\sum^D (z - \hat{z})^2}{D}} \tag{18}$$

Based on the empirical distribution of NRMSD values across all the present-day human populations for which a fit was obtained, we use a cut-off of NRMSD = 0.29 to identify reliable exponential fits (Notes S1.3 in S1 Text), though in practice other thresholds may also be valid.

We define a founder event to be <u>significant</u> if the following criteria were met: (i) the 95% confidence intervals of the estimated founder age and intensity do not include 0, (ii) the estimated founder age is lower than 200 generations and its associated standard error is lower than 50 generations, (iii) the estimated founder intensity is greater than 0.5%, and (iv) the NRMSD is lower than 0.29. These thresholds are based on simulation results (see Note S1 and S2 in S1 Text for details).

## Software availability

*ASCEND* is written in Python and is available for download (with tutorial and examples) at: https://github.com/sunyatin/ASCEND. This includes both the Naïve and FFT implementations.

## Simulations

In order to investigate the accuracy of *ASCEND* in estimating the age and intensity of the founder event, we performed simulations under a range of demographic models using the coalescent simulator *msprime* [76]. All simulations involved at least two populations (one target, *A* and one outgroup, *O*) which diverged 1,800 generations ago (Fig 1). Unless stated otherwise, for each simulation and each population, we generated data for 30 haploid individuals for a total of 20 chromosomes of size 50 megabases each. We assumed a mutation rate of $1.2 \times 10^{-8}$ per base pair per generation [77] and a recombination rate of $1 \times 10^{-8}$ per base pair per generation [78]. Except during the founder event, the two populations had an effective population size of 12,500. For simulations of admixture followed by a founder event, population *A* admixed 110 generations before present with 60% ancestry from a population *A'* and 40% ancestry from a population *B'* (*A'* and *B'* diverged 1,800 generations ago). We combined two haploid chromosomes at random without replacement to generate the diploid genotypes. We applied *ASCEND* to each simulation and reported the mean and the standard error of the parameters based on a chromosome jackknife, where we drop one

chromosome in each run. For some scenarios such as the single-generation bottleneck model, we generated data for 10 replicates and assessed the variation across runs. We found that the inferred estimates of founder age and intensity were qualitatively similar across replicates (within overlapping 95% confidence intervals) (Tables A and B in S1 Text). Details are described in Notes S2 in S1 Text.

## Datasets

**Human datasets.** We applied *ASCEND* to samples from worldwide present and ancient human populations from three datasets: (i) Human Origins version 37.2 dataset (HO37, release of the Allen Ancient DNA Resource, AADR) for present-day populations: we used 5,637 present-day individuals from 530 groups restricting to data from 499,158 autosomal SNP positions genotyped using the Affymetrix Human Origins array; (ii) IndiaHO: this dataset includes 1,662 individuals from 249 ethno-linguistic groups genotyped on the Affymetrix Human Origins array [25]; (iii) Human Origins version 44.3 (HO44, AADR release v44.3) for ancient populations: this dataset comprises 5,225 ancient individuals from 1,800 groups, restricting to data from 1,233,013 autosomal SNP positions that have been genotyped using the Affymetrix Human Origins array. We used the genetic positions from the 1000 Genomes recombination map [79].

For all three datasets, we limited our analysis to groups with a minimum of 5 samples. We removed individuals that were marked as duplicates, low coverage or outliers in the original studies. We filtered the datasets to remove any potential close relatives defined as: (i) pairwise genomic sharing ($\pi$) greater than 0.45 with any other individual, or (ii) having both $\pi > 0.125$ (third-degree relatives) and at least one IBD segment greater than 65 cM (on average, half the chromosome). For identification of IBD segments, we first phased the samples from each dataset using EAGLE 2.4.1 with default parameters [80], using the 1000 Genomes Project phased samples as a reference panel to increase the phasing accuracy [79]. The IBD segments were then called using GERMLINE 1.5.3 [81] with default parameters (*-bits 75, -err_hom 0, -err_het 0, -min_m 3*) in the *-genotype* extension mode. Note, $\pi$ is an estimator of the proportion of genome-wide IBD (for example, $\pi = ½$ for first-degree relatives). We computed pairwise $\pi$ using PLINK v1.90b6.2 *genome* module [82]. In the case of ancient DNA, since the genotypes are pseudo-haploid, we could not phase haplotypes and therefore could not detect IBD segments, therefore we only used the criterion of $\pi > 0.125$ to identify close relatives. We applied the HaploScore algorithm to remove false positive IBD segments using the recommended parameters, namely genotype error of 0.75%, switch error of 0.3% and the threshold matrix for a mean overlap of 80% [83]. For details about the data curation, see Notes S3 in S1 Text.

**Dog datasets.** We downloaded two publicly available genetic datasets for dogs: (i) Sams dataset: this includes 1,792 individuals from 11 breeds genotyped on 175,123 SNPs (https://figshare.com/articles/dataset/Supplementary_Material_for_Sams_and_Boyko_2018/7330151) [50], and (ii) Hayward dataset: this contains 4,342 individuals from 198 breeds genotyped on 160,723 SNPs (https://datadryad.org/stash/dataset/doi:10.5061/dryad.266k4) [51]. We used the genetic positions from the CanFam3.1 genetic map [84].

For both datasets, we filtered out SNPs and individuals with missingness greater than 1% and 5% respectively. To remove close relatives, we computed $\pi$ between all pairs of individuals using PLINK v1.90b6.2 *genome* module [82]. For each pair of individuals with $\pi$ greater than 0.45, we excluded one of the individuals out of the pair. Compared to the human datasets, we did not include the filter criterion based on the IBD segments as phasing would be less reliable with small sample size and without the use of a reference [85]. Instead, to control for inbreeding, we excluded any individual with at least one run of homozygosity (ROH) longer than 30

cM (on average, half the chromosome). To compute ROH, we used the PLINK *homozyg* module. We limited our analysis to all groups with a minimum of 5 samples.

## Supporting information

**S1 Fig. Decay curves for present-day human populations from the HO37 and IndiaHO datasets.** The *X*-axis represents the genetic distance (in cM) and the *Y*-axis represents the average allele sharing correlation (or weighted allele sharing covariance for populations with pseudo-haploid data, cf. S1 Table). The legend shows the mean and 95% confidence interval for the founder age ($T_f$) in generations before present (gBP) and the founder intensity ($I_f$), as well as the NRMSD (see Methods). The panels are grayed when the exponential fitting failed or when the evidence for the founder event was not significant (see Methods). The specific reason is highlighted in red in the legend.
(PDF)

**S2 Fig. Distribution of the founder age and intensity estimated in present-day Jewish communities from the HO37 dataset.** Panel (A) shows the distribution of founder intensity, and panel (B) shows the distribution of founder ages for all Jewish populations that passed our filtering criteria in the HO37 dataset and showed evidence for a significant founder event (see Methods). The populations are ordered by decreasing order of founder intensity, from top to bottom. (*A) Distribution of the estimated founder intensities*. We show the mean founder intensities (points) and their associated 95% confidence intervals. (*B) Distribution of the estimated founder ages*. We show the estimated founder ages and their associated 95% confidence intervals. The estimated ages were converted from generations to years before present by using a generation time of 28 years [29,30].
(PDF)

**S3 Fig. History of founder events in present-day South Asian populations from the IndiaHO dataset.** Panel (A) shows the distribution of founder intensity and panel (B) shows the distribution of founder ages for all present-day populations that passed our filtering criteria and showed evidence for a significant founder event in the IndiaHO dataset (see Methods). Each point represents a population and the shape of the points indicates whether the population lives on an island (triangle) or land (circular). The colors represent the linguistic affiliation of the groups. The populations are ordered, first by their linguistic affiliation and then, by decreasing order of estimated founder intensity, from top to bottom. (*A) Distribution of the estimated founder intensities*. We show the founder intensities (points) and their associated 95% confidence intervals. The black horizontal line shows the estimated founder intensity in the Ashkenazi Jew population from the IndiaHO dataset (2.0%). (*B) Distribution of the estimated founder ages*. We show the estimated founder ages and their associated 95% confidence intervals. The estimated ages were converted from generations to years assuming an average generation time of 28 years [29,30].
(PDF)

**S4 Fig. Decay curves for ancient human genomes from the HO44 datasets.** The *X*-axis represents the genetic distance (in cM) and the *Y*-axis represents the average weighted allele sharing covariance. The legend shows the mean and 95% confidence interval for the founder age ($T_f$) in generations before the sampling age of the ancient specimens (gBS), the founder intensity ($I_f$), as well as the NRMSD (see Methods). The panels are grayed when the exponential fitting failed or when the evidence for the founder event was not significant (see Methods, here

using the standard maximum founder age of 200 gBS). The specific reason is highlighted in red in the legend.
(PDF)

**S5 Fig. History of founder events in ancient human populations from the Human Origins v44 (HO44) dataset.** Panel (A) shows the distribution of founder intensity, and panel (B) shows the distribution of founder ages for all ancient populations that passed our filtering criteria in the HO44 dataset and showed evidence for a significant founder event (see Methods). Each point represents a population and the color of the points indicates the geographical (continent) location of the population. The populations are ordered, first by their continent and then, in increasing order of radiocarbon sample age, from top to bottom. *(A) Distribution of the estimated founder intensities*. We show the mean founder intensities (points) and their associated 95% confidence intervals. The black horizontal line shows the inferred founder intensity in the present-day Ashkenazi Jewish population from the HO37 dataset (1.7%, CI95: [1.3%–2.1%]). *(B) Distribution of the estimated founder ages*. We show the estimated founder ages and their associated 95% confidence intervals. The estimated ages were converted from generations to years before present by using a generation time of 28 years [29,30] and by adding the radiocarbon sample date (black diamond-shaped point) of the specimens.
(PDF)

**S6 Fig. Decay curves for all dog breeds surveyed in the Sams and Hayward datasets.** The *X*-axis represents the genetic distance (in cM) and the *Y*-axis represents the average allele sharing correlation. The legend shows the mean and 95% confidence interval for the founder age ($T_f$) and the founder intensity ($I_f$), as well as the NRMSD (see Methods). The panels are grayed when the exponential fitting failed or when the evidence for the founder event was not significant (see Methods). The specific reason is highlighted in red in the legend.
(PDF)

**S7 Fig. Temporal patterns of founder events in human history and pre-history.** We represented the distribution of the radiocarbon ages for the surveyed ancient populations from the HO44 dataset (gray histogram) as well as the distribution of the founder ages estimated across populations with evidence of founder event (yellow histogram). The *X*-axis is in years BP. The founder ages reported in the plot were converted from generations to years before present by using a generation time of 28 years [29,30]) and by adding the radiocarbon sample date. Despite a higher population sampling towards the present (left), we observe a shift of the founder age distribution towards the more distant past (right).
(PDF)

**S1 Table. Data curation for all present-day and ancient human populations from the HO37, IndiaHO and HO44 datasets.**
(XLSX)

**S2 Table. Founder event estimates for all present-day and ancient human populations from the HO37, IndiaHO and HO44 datasets.**
(XLSX)

**S3 Table. Comparison of founder event ages estimated in present-day human populations from the HO37 dataset with published dates of admixture.**
(XLSX)

**S4 Table. Data curation for dog breeds from the Sams and Hayward datasets.**
(XLSX)

**S5 Table. Founder parameter estimates for dog breeds from the Sams and Hayward datasets.**
(XLSX)

**S6 Table. Comparison of the founder parameter estimates for dog breeds between the Sams and Hayward datasets.**
(XLSX)

**S1 Text. Detailed description of the method, simulation studies, data curation and additional analyses.**
(PDF)

## Acknowledgments

We thank Monty Slatkin, Shai Carmi, Shamam Waldman, Mathieu Gautier, and members of the Moorjani lab for helpful discussions. We thank Manjusha Chintalapati for information about the group affiliation of ancient DNA individuals, and Kumarasamy Thangaraj and Pratheusa Machha for information about the group affiliation of present-day South Asian populations. We thank Monty Slatkin, Nick Patterson, Manjusha Chintalapati, Sonal Singhal, Kelsey Witt, Shai Carmi and Emilia Huerta-Sanchez for helpful comments on the manuscript. We thank Indro Fedrigo for his help with high performance computing resources.

## Author Contributions

**Conceptualization:** Rémi Tournebize, Priya Moorjani.

**Data curation:** Rémi Tournebize, Priya Moorjani.

**Formal analysis:** Rémi Tournebize.

**Funding acquisition:** Priya Moorjani.

**Investigation:** Rémi Tournebize, Priya Moorjani.

**Methodology:** Rémi Tournebize, Priya Moorjani.

**Project administration:** Priya Moorjani.

**Software:** Rémi Tournebize, Gillian Chu.

**Supervision:** Priya Moorjani.

**Validation:** Priya Moorjani.

**Visualization:** Rémi Tournebize.

**Writing – original draft:** Rémi Tournebize, Priya Moorjani.

**Writing – review & editing:** Rémi Tournebize, Priya Moorjani.

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
