## [Decision Letter · Decision Letter 0]

13 Mar 2022

Dear Dr Tournebize,

Thank you very much for submitting your Research Article entitled 'Reconstructing the history of founder events using genome-wide patterns of allele sharing across individuals' to PLOS Genetics.

The manuscript was fully evaluated at the editorial level and by independent peer reviewers. The reviewers appreciated the attention to an important topic but identified some concerns that we ask you address in a revised manuscript

We therefore ask you to modify the manuscript according to the review recommendations. Your revisions should address the specific points made by each reviewer.

[LINK]

Yours sincerely,

Kirk E Lohmueller

Guest Editor

PLOS Genetics

David Balding

Section Editor: Methods

PLOS Genetics

Editor Comments:

Thank you for submitting your manuscript to PLoS Genetics. It has now been evaluated by three reviewers. All three were quite positive about ASCEND and its potential impact to the field. Further, all were impressed by the detailed simulation studies performed to assess the performance of the method. I concur with their assessment. While the reviewers were generally positive, they had a number of suggestions for additional simulations and clarifications to be made in the text. Reviewer 3 suggests performing simulations without a founder event and assessing the performance of ASCEND. Reviewer 2 suggests examining performance of ASCEND when there has been admixture more recently than the founder event. I think both of these situations are important to investigate with additional simulations. I do not think that you need to consider many scenarios for each of these types of models. Rather, just one or two cases that you think are reasonable should suffice. Please address the other more minor comments from all three reviewers as well. I look forward to receiving your revised manuscript.

Reviewer's **Comments:**

Reviewer #1: See attached document

Reviewer #2: Title: Reconstructing the history of founder events using genome-wide patterns of allele sharing across individuals. Authors: Tournebize et al., 2022

Summary: Propose a new method called ASCEND that quantifies correlation of shared alleles between individuals to infer the age and strength of founder events, apply to several large datasets (humans and canids) to infer recent founder events. The algorithm itself has three steps: (1) at each marker, infer the total number of IBS alleles shared across pairs of individuals; (2) compute correlation in shared allele counts across individuals in a population; (3) measure decay of this correlation with genetic distance; expected to have exponential decay with distance, amplitude of decay is ~ strength of bottleneck.

I have reviewed the math and it’s well described and elegant - I have to congratulate the authors on this massive work comprising both simulations and analyses of empirical data.

Claims: 1) Number, 2) Strength, 3) tempo of bottleneck events over history, 4) timing of bottleneck event (founder age).

The proposed approach addresses the strength, and the timing of most recent bottleneck event, and not the number/tempo/strength of multiple serial bottleneck events. I do like that the authors don’t overplay this in the later results (e.g. testing with a model of two epochs).

Important - comparison of methods perhaps? This would be important for benchmarking - if computational constraints make this intractable (e.g. size of datasets, availability of reliable phase information, linkage maps), perhaps at least comparison of estimated demographic parameters such as the founder size/age would be useful for users.

Resolved - these are very well detailed in the Supplementary material.

In step 2, the assumption here is that individuals that share IBS are IBD? However, IBS could be very well due to population structure? How does ASCEND account for this?

--Maybe addressed by removing closely related individuals in their human analyses? But how do the authors propose that this is achieved for datasets without pedigree information?

They state that they compute a cross population correlation to account for ancestral allele sharing - but this also is confounded if there’s recent migration - perhaps addressed by the second model of recent gene flow tested (Fig. S2.4.2)?

I see that this model is one of admixture prior to the bottleneck event, and not after the bottleneck event; post admixture decay of IBD segments will be accelerated during a bottleneck and therefore would not be expected to affect the number of shared alleles. However, if the admixture event occurs post a bottleneck, as it does with most human populations (e.g. recent admixture in Europeans, or recent admixture in North Africans), IBS is definitely != IBD. I think that this study would benefit from one more simulation study, looking at models of recent admixture (post bottleneck), and assess performance of ASCEND on such data.

I also found the GitHub page for ASCEND to be very detailed. I do think that users will benefit from a vignette/tutorial of some sort though, perhaps even using their example dataset.

Reviewer #3: Tournebize et al present ASCEND - a new method to infer the timing and strength of founder (e.g. bottleneck) events from genotype data. ASCEND computes the correlation in pairwise allele sharing between pairs of sites separated by varying genetic distances and relates this to founder events. It fits a exponential decay of this correlation with distance, to me this is broadly similar to the ALDER/ROLLOFF methods to infer admixture histories, but with different statistics and interpretation. Briefly, the correlation at the smallest genetic distances considered reflects the intensity of the founder event, and the rate of decay in correlation with genetic distance reflects the timing of the founder event.

The manuscript contains quite a number of simulations and analyses, with 6 different simulation scenarios and 184 human populations and 40 dog breeds going . The results are presented in detail in the supplement, and summarized in the main text. Broadly, I found these analyses were conducted with care and an understanding of the many potential pitfalls that can arise from the analysis of large-scale genetic data.

Based on the simulated data, the authors report the ability to infer the timing and intensity of founder events for events with 200 generations, and show the method is decently robust to a range of demographic scenarios containing a founder event.

By applying the method to real data sets from humans and dogs, the authors present evidence for founder events in the history of many populations, with results largely consistent with prior studies. However, one of the main advantages of ASCEND vs competing methods is the reduced data requirements. ASCEND requires many fewer samples, doesn’t require phasing, and can be applied to data sets with missing data or haploid samples - the authors do recommend access to genetic data from an outgroup population not subject to the founder event though - which may be difficult in some cases. For comparison, competing methods (e.g. IBDNe) require access to hundreds of samples and require phased data. However one possible catch is that the authors suggest caution when interpreting results derived from what seem like somewhat common scenarios, e.g. exponential growth following the founder event.

I was able to download ASCEND from the github repo and run an analysis on the example data set. Everything seemed to run fine, after a small bit of debugging due to error when running on compressed data files. I would encourage the authors to provide version information for the required packages, as I received a number of depreciation warnings when running the code.

Large issues:

Broadly, I feel that authors did a very good job of trying to consider how different demographic scenarios in combination with a founder event could affect the analysis, with the authors testing 6 total scenarios. However, I think there is an important aspect missing here - consideration of false positives. One of the main claims of this manuscript is the ability to detect (and characterize) founder events. The authors go into some detail evaluating how well existing founder events are characterized, but never evaluate the application of ASCEND to data from populations lacking founder events. This is important because researchers want to be relatively sure that when ASCEND reports a founder event, there was one. This could be done by applying ASCEND to simple demographic models with constant / growing populations without founder events, and or maybe something like the 3-population human out of africa model. This more complicated model does have founder events and population growth, but the founder events seem to be older than the focus of ASCEND.

Small issues:

Overall, I found the writing clear, but I suggest the notation should be adjusted to improve clarity. These are not large changes, but can aid in reader comprehension

S_d vs S(d) on line 615

d in units of Morgans (line 612) or cM (line 718)

i indexing “SNPs” (line 702) or individual (line 609)

Use of the all the terms “marker” “SNP” “locus”, “variant site” within the methods- do these all refer to the same thing or are there meaningful differences?

Line 27: In these few sentences both the terms “human populations” and “surveyed groups” are used, but the distinction is not fully clear.

Line 52 - It’s not clear to me what “diverse” means here - genetically diverse or something else?

Line 84 - clarify that this is given in generations, not years.

Line 150 - I am a bit confused about the units for intensity of a founder event. It is often shown in this paper as a percent, but a percent of what? (generations per ind?). For clarity, I suggest presenting it as a decimal, as it appears in your citation #9.

Line 553 - missing “of”

Line 568 - ROH is not defined prior to this.

Line 672 - Make it more clear you assume no recombination during the bottleneck.

Line 683 - I was a bit confused by this sentence, also is there a citation for the relation between z and expected time to coalescence?

Figure 1

Figure 1. I found the resolution a bit lacking, otherwise a nice figure. In panel C, I noticed that the true founder age was not evaluated on the same range of values as in panel B, but it was not clear why.

Figure 2. I think the current figure is a nice compromise between visual appeal and data presentation, but it could be improved by the reduction of overplotting, perhaps by adjusting the spacing of the data points.

Figure 3 - The violin plot outlines add color, but it is unclear what the relative widths of the violins represent within and between plots - do they all contain the same area?

Figure 4

Denoting the breeds / roles solely via pictograms might be difficult for readers not familiar with dogs. The images are great, but I suggest they are accompanied by text labels. Also, I might suggest that the results be presented in a more genetic / evolutionary context by organizing the dog breeds via an evolutionary tree, rather than in a circle plot organized by founder intensity

Figure 5 - Great figure, I used it often to help interpret the results and text. I might

suggest putting this figure first and removing the current Figure 1A.

**Have all data underlying the figures and results presented in the manuscript been provided?**

Reviewer #1: Yes

Reviewer #2: Yes

Reviewer #3: Yes

PLOS authors have the option to publish the peer review history of their article (what does this mean?). If published, this will include your full peer review and any attached files.

Reviewer #1: No

Reviewer #2: **Yes: **Arun Sethuraman

Reviewer #3: No

---

## [Decision Letter · Decision Letter 1]

8 May 2022

Dear Dr Tournebize,

We are pleased to inform you that your manuscript entitled "Reconstructing the history of founder events using genome-wide patterns of allele sharing across individuals" has been editorially accepted for publication in PLOS Genetics. Congratulations!

Yours sincerely,

Kirk E Lohmueller

Guest Editor

PLOS Genetics

David Balding

Section Editor: Methods

PLOS Genetics

Comments from the reviewers**:**

Reviewer #1: The authors did an excellent job thoughtfully responding to each of my comments. I have enjoyed reading both the original and revised version of their manuscript.

Reviewer #2: Dear Authors,

Thank you for carefully addressing my previous concerns regarding scenarios of recent admixture - I think that this revision sufficiently captures a comprehensive analysis of ASCEND's utility. I commend you on this massive work!

Best,

Arun Sethuraman

Reviewer #3: The authors have thoroughly responded to and addressed all the issues raised during the first review round.

**Have all data underlying the figures and results presented in the manuscript been provided?**

Reviewer #1: Yes

Reviewer #2: Yes

Reviewer #3: Yes

PLOS authors have the option to publish the peer review history of their article (what does this mean?). If published, this will include your full peer review and any attached files.

Reviewer #1: No

Reviewer #2: **Yes: **Arun Sethuraman

Reviewer #3: No

**Data Deposition**

http://datadryad.org/submit?journalID=pgenetics&manu=PGENETICS-D-22-00145R1

**Press Queries**

---

## [Editor Report · Acceptance letter]

3 Jun 2022

PGENETICS-D-22-00145R1 

Reconstructing the history of founder events using genome-wide patterns of allele sharing across individuals 

Dear Dr Tournebize, 

We are pleased to inform you that your manuscript entitled "Reconstructing the history of founder events using genome-wide patterns of allele sharing across individuals" has been formally accepted for publication in PLOS Genetics! Your manuscript is now with our production department and you will be notified of the publication date in due course.

With kind regards,

Olena Szabo

PLOS Genetics

On behalf of:
